# Links between problem gambling and spending on booster packs in collectible card games: A conceptual replication of research on loot boxes

**David Zendle** [1] *, **Lukasz Walasek**[2], **Paul Cairns**[1], **Rachel Meyer**[1], **Aaron Drummond**[3]

**1** Department of Computer Science, University of York, York, United Kingdom, **2** Department of Psychology, University of Warwick, York, United Kingdom, **3** School of Psychology, Massey University, Palmerston North, New Zealand

* david.zendle@york.ac.uk

**Data Availability Statement:** The data and R script underlying this study are available on OSF (https://osf.io/abx2t/).

## Abstract

Loot boxes are digital containers of randomised rewards present in some video games which are often purchasable for real world money. Recently, concerns have been raised that loot boxes might approximate traditional gambling activities, and that people with gambling problems have been shown to spend more on loot boxes than peers without gambling problems. Some argue that the regulation of loot boxes as gambling-like mechanics is inappropriate because similar activities which also bear striking similarities to traditional forms of gambling, such as collectable card games, are not subject to such regulations. Players of collectible card games often buy sealed physical packs of cards, and these 'booster packs' share many formal similarities with loot boxes. However, not everything which appears similar to gambling requires regulation. Here, in a large sample of collectible card game players (n = 726), we show no statistically significant link between in real-world store spending on physical booster and problem gambling (p = 0.110, $\eta^2$ = 0.004), and a trivial in magnitude relationship between spending on booster packs in online stores and problem gambling (p = 0.035, $\eta^2$ = 0.008). Follow-up equivalence tests using the TOST procedure rejected the hypothesis that either of these effects was of practical importance ($\eta^2$ > 0.04). Thus, although collectable card game booster packs, like loot boxes, share structural similarities with gambling, it appears that they may not be linked to problem gambling in the same way as loot boxes. We discuss potential reasons for these differences. Decisions regarding regulation of activities which share structural features with traditional forms of gambling should be made on the basis of definitional criteria as well as whether people with gambling problems purchase such items at a higher rate than peers with no gambling problems. Our research suggests that there is currently little evidence to support the regulation of collectable card games.

**Funding:** PC receives buyout from the EPSRC Doctoral Centre for Intelligent Games and Games Intelligence. AD is supported by the Marsden Fund Council from NZ Government funding, managed by Royal Society Te Apārangi; MAU1804. The funders had no role in study design, data collection and analysis, decision to publish, or preparation of the manuscript.

**Competing interests:** The authors have declared that no competing interests exist.

## Introduction

Recently, there has been a surge in public policy interest in the incorporation of gambling-like mechanics in video games [1,2]. In particular, the concerns have focused upon loot boxes–digital containers of randomised rewards often purchasable for real world money. Many loot boxes appear to meet the psychological (Drummond & Sauer, 2018), and legal [3] definitional characteristics of traditional gambling practices. As such, many policymakers have regulated (e.g., Belgium, Japan, the Netherlands) or considered regulating (e.g., Australia, New Zealand, UK, US) loot boxes in some way. One argument that has been put forward by industry representatives is that many existing activities, such as Collectable Card Games (CCGs), utilise the same mechanics as loot boxes and are not currently subject to such regulation [4]. Researchers have certainly raised concerns about the possibility that CCGs might constitute a form of gambling (e.g., [5]). However, just because something appears to resemble gambling does not necessarily mean it is a form of gambling. Chief among the key differences between the case for regulating loot boxes and the case for regulating CCGs is the mounting evidence that players with higher problem gambling symptomology spend more on loot boxes than players with fewer problem gambling symptoms, suggesting that people with gambling problems may engage with them in a similar manner to traditional gambling activities [2,6–13]. No such evidence currently exists for CCGs. Here, we investigate whether people with higher problem gambling symptomology also spend more on CCGs than people with fewer problem gambling symptoms. If they do, this may imply that a discussion about whether CCGs require regulation is needed. If they are not purchased disproportionately by people with gambling problems, then this may imply that CCGs are not engaged with in a similar manner to traditional forms of gambling, and that regulators should not treat CCGs as equivalent to loot boxes.

### What are loot boxes?

Loot boxes are items in video games that may be bought for real-world money but contain randomised contents. The value of any given loot boxes content is also uncertain at the time of purchase. For example, players of the football game *FIFA Ultimate Team* may pay real-world money to purchase digital 'player packs' in-game. These virtual packs contain new footballers that gamers may use when playing *FIFA*. The contents of a player pack may be both desirable and valuable–or they may not be. One may pay to open a pack and receive a high-powered and helpful player such as Arsenal's Pierre-Emerick Aubameyang. Or one may instead receive a less powerful and useful player such as Oxford United's Shandon Baptiste.

Loot box mechanics are prevalent on both mobile and desktop games. Indeed, the majority of top-grossing mobile games appear to contain loot boxes [14]. Furthermore, rates of exposure to loot boxes appear to have been high for several years. Analysis of publicly-available daily player counts from the online gaming platform *Steam* suggests that the majority of desktop play sessions have taken place in games that feature loot boxes since as early as 2015 [15].

### Loot boxes and gambling

The randomisation of rewards present in loot boxes has led to comparisons between them and gambling. Consider placing a bet on a roulette wheel and opening a loot box. In both cases, individuals are staking real-world money on the chance outcome of a future event. These similarities have led to suggestions that some loot boxes may be "psychologically akin" to gambling [1]. These similarities might result in greater engagement in gambling amongst vulnerable groups, and in turn, lead to the development of problem gambling amongst this group [16]. Problem gambling is commonly considered to be extremely harmful. It refers to an excessive and disordered engagement with gambling activities that is typically outside of the gambler's

volitional control and leads to severe problems in their personal, financial, and professional lives [17,18].

Recent research has established the existence of a reliable link between loot box spending and problem gambling: The more that a gamer spends on loot boxes, the more they also appear to be at risk of gambling problems [6,8,9,19,20]. It is unclear what this link represents. It may, as suggested above, represent a situation in which engagement with loot boxes literally causes problem gambling. Alternatively, it may be that people with gambling problems find loot boxes particularly attractive, and spend disproportionally more money on them than other players. However, in either case, academics have argued that links between problem gambling and loot box spending may be an important potential source of real-world harm [7,10,12,21].

## What are booster packs?

In collectible card games players are often able to buy sealed physical packs of cards, known as booster packs. The contents of booster packs are typically determined in a random manner. Thus, the value of any given booster pack is of uncertain value to the player at the point of purchase, and they are thus similar to loot boxes in video games. For example, players of the strategic battle card game *Magic*: *The Gathering* may spend money to purchase packs containing additional cards to use when playing the game. The contents of these packs vary in desirability and value. A player may open a booster pack and receive a *Mox Opal* card, for example, which is extremely useful and hence can fetch more than $100 on a resale market. However, sometimes the contents of a booster pack are neither desirable nor valuable. Players may open a booster pack and merely receive cards like *Memoricide*, *Steel Hellkite*, and *Tunnel Ignus*, which are less useful and fetch only a few cents on resale markets. When purchasing booster packs, players of collectible card games therefore cannot know the value of what they will receive in return for their money.

**Gambling and booster packs.** In [22], Griffiths defines a set of criteria by which gambling activities may be differentiated from other forms of behaviour:

1. Gambling involves an exchange of money or other valuable goods

2. This exchange is determined by a future event, whose outcome is unknown when the bet is made

3. This event's result is determined (at least in part) by chance

4. Winners gain at the sole expense of losers

5. Losses can be avoided by not taking part in the gamble

Drummond and Sauer [1] note that a number of loot boxes appear to meet these criteria, and hence could be considered psychologically akin to gambling. By extension, these criteria may be also seen to apply to booster packs in CCGs. When purchasing a booster pack, players are involved in an exchange of money or valuable goods: A booster pack for the *Pokemon* collectible card game, for example, currently costs approximately £3.99 and contains 11 randomly-assorted cards. Similarly, the success of this exchange is determined by a future event (the opening of the booster) whose outcome is unknown when the booster is purchased. Drummond and Sauer argue that just as with loot boxes, winners gain at the expense of losers in that those who receive useful cards in their booster packs gain an in-game competitive advantage over those who do not receive useful and rare cards. Finally, a loss of money can be avoided simply by not buying a booster pack.

Indeed, some similarities between physical card collections and digital loot boxes are so great that some video game loot boxes take the appearance of 'virtual' card packs. For example, *FIFA Ultimate Team's* 'player packs' are presented in-game in a way that mirrors the opening of physical packs of playing cards. Even more strikingly, some video games adopt overarching in-game metaphors that directly mimic traditional physical collectible card games, and incorporate loot boxes into these games in the form of digital card packs. *Hearthstone*, for example, is a computer game in which players collect and battle using virtual cards in a similar manner to physical card games such as *Magic*: *The Gathering* and the *Pokemon* collectible card games. Players of *Hearthstone* may pay real world money to purchase virtual 'packs' of digital cards. However, crucially, this monetisation technique involves disbursing virtual and not physical goods, and takes place within the context of a video game: whilst it may have graphics which seek to mimic real-world booster packs, it nonetheless constitutes a video game loot box under out definitions, due to its virtual content and presence within a video game rather than a booster pack which distributes physical cards for a physical card game.

The similarities outlined above have led to comparisons between collectible card games and loot boxes by members of the video game industry [23]. These similarities have led to significant policy interest in whether the effects of loot boxes and booster packs are fundamentally equivalent: The terms of reference of a recent evidence call by the UK government, for example, requested information regarding "Whether any harms identified [to be associated with loot boxes] also apply to offline equivalents of chance mechanisms, such as buying packs of trading cards" [24].

**Differences between booster packs and loot boxes.** Booster packs therefore share many formal similarities with both loot boxes and more traditional gambling activities. However, it is important to note that loot boxes and booster packs, whilst similar, are not identical. A key distinction between booster packs and loot boxes lies in the fact that loot boxes are purchased and opened in an entirely virtual environment. There is no such thing as a physical player pack for *FIFA* or a physical weapon case for *Counter-Strike*: *Global Offensive*. Both payment for these items, acquisition of these items, and the opening of these items take place online in a digitally-mediated fashion. By contrast, the opening of booster packs necessarily takes place in a physical space. Whilst booster packs may be bought either in a physical store or via an online storefront, they must always be opened in the real world. Because of this qualitative difference between loot boxes and physical collectible card games, there may be important real-world differences in the experience and effects of engagement with these items.

First, the digital nature of loot boxes means that companies can present the opening of them in creative ways that would not be possible in the real world. Often, these methods are designed to increase the engagement of the player, using a range of salient visual and auditory cues. For example, when opening loot boxes in *Counter-Strike*: *Global Offensive*, players are presented with a slot machine-like interface which displays to them a revolving reel of potential prizes, eventually settling on the item which is inside the box that they are opening. When opening a player pack in *FIFA*, the in-game interface is similarly exciting in nature, and presents players with a spectacular light-show if a rare player is uncovered during an opening [25]. By contrast, opening a physical booster pack involves tearing open a physical pack of cards–whilst such packs may contain exciting foil covers, the audio-visual experience associated with this event is much more constrained.

The importance of visual and auditory cues in promotion of gambling products has been extensively studied. Both types of cues contribute to the subjective arousal experienced by a gambler, in addition to the arousal elicited by the prospect of uncertain gains and losses [26,27]. In fact, overlapping neural substrates responsible for processing rewards, emotional responses, and habit formation appear to be reactive to cues (e.g. visual) associated with

gambling [28,29]. Casinos appear to strategically choose the background music and sounds produced by the gambling machines to encourage the development and maintenance of gambling behaviour [30]. It is therefore possible that the visual and auditory cues associated with loot box openings contribute to their appeal for people with gambling problems.

Second, the digital nature of loot boxes allows players to buy, acquire, and open these products very rapidly. Videos posted to video streaming sites like *Youtube*, for example, demonstrate that games are set up to allow gamers to purchase, receive, and open thousands of dollars' worth of loot boxes over just a few minutes time [31,32]. By contrast, booster packs take more time to be opened. It is necessarily slower to go to a storefront, find the booster pack for the game of one's choice, purchase said booster pack, find a suitable environment to open it in, and then open it. By comparison, the environment in which a loot box is opened is custom-built to ensure that the experience of spending is as rapid and smooth as possible. In recognition of the danger posed by the possibility of gambling away ones money very rapidly, recent gambling regulation in the UK imposed the £2 limits on fixed odds betting machines [33].

From the perspective of behavioural science, the ease with which loot boxes are purchased and opened is particularly relevant to problem gambling. It is well known that when facing a choice between amounts of money that could be received at different points in time, people typically demand a premium to accept a reward that will be received in a more distant future. In other words, people discount future rewards and are willing to accept less money with the immediate effect. In numerous studies, steeper discount rates of future rewards have been associated with various forms of addiction (alcohol, cigarettes, cocaine) as well as financial mismanagement [34]. People with gambling problems also appear to be more impulsive and have higher discounting rates of future rewards than non-gamblers, which suggest that more rapid discounting of future rewards may be a route to problem gambling [35,36]. The instantaneous nature of loot boxes in today's video games seems to lift barriers that could otherwise prevent those at risk of developing a gambling problem from engaging in games of chance.

Furthermore, it is important to note that the way that cards are played and collected amongst players of physical collectible card games may differ from how loot box contents are accrued by players of video games. For instance, anecdotal reports suggest that players of collectible card games may end up owning large number of cards through means other than buying. For example, the tangible nature of physical card games mean that players may gift or trade cards with each other. In a video game, rules for the transfer of property between players are controlled by the architects of the game, and may not allow such transfers to take place. For example, loot box contents in the video game *Last Shelter*: *Survival* are locked to a player's account, and may not be traded between players [37]. Conversely, winning packs, cards, or decks by participating in formal and informal tournaments is thought to be a common occurrence amongst players of physical card games. Such bonuses may not translate to all loot box implementations.

Finally, it is important to note that specific technological differences between loot boxes and booster packs may be responsible for different relationships between these things and problem gambling. For example, many video games do not disclose the odds of loot boxes, and sometimes even algorithmically adjust the odds of receiving different rewards as people spend money [37]: This kind of manipulation is not possible when buying booster packs. In a similar vein, aggressive or otherwise misleading advertising regarding microtransactions are reported by players of video games [38]: Such advertising may differ in its form and prevalence within the context of physical collectible card games.

**Summary.**   On the face of it, there appear to be both substantial similarities and substantial differences between CCG booster packs and loot boxes. However, there is no evidence

regarding whether the correlates of booster pack spending mirror the correlates of loot box spending: while people with gambling problems have repeatedly been shown to spend more on loot boxes than their peers without gambling problems, presently no such evidence exists for CCGs.

Some stakeholders argue that loot boxes and booster packs are fundamentally equivalent activities. If this were the case, one would expect that effects observed in the context of loot boxes may replicate in the context of booster packs. However, another perspective might suggest that fundamental differences between loot boxes and booster packs mean that their correlates (and potential effects) differ in important ways.

In short, although CCGs appear to be similar to loot boxes, it is important to establish whether they are associated with problem gambling symptomology in the same way as loot boxes. If no link between problem gambling symptomology and CCGs exist, then narratives regarding the identicality of loot boxes and CCGs may be unhelpful to understanding the effects of loot boxes. Here, we investigate both the size and importance of links between spending on booster packs and problem gambling to provide empirical evidence on whether problem gambling symptomology is linked to spending on CCG booster packs.

**Hypotheses.**   Based on the fact that CCG booster packs structurally resemble loot boxes, we predicted that there would be a relationship between problem gambling symptoms and CCG booster pack purchasing (**H1**-**H2**). We also investigated (**H3**-**H4**) whether the relationship between problem gambling and spending on booster packs was greater than or less than an $\eta^2$ of 0.04, which is commonly defined as being a threshold for effects of clinical significance [39]. Specifically, we predicted

H1.   The more money an individual spends on booster packs in real-world stores, the more severe their problem gambling is.

H2.   The more money an individual spends on booster packs in digital stores, the more severe their problem gambling is.

H3.   The strength of the relationship between spending on physical booster packs in real-world stores and problem gambling will be equivalent to magnitude $\eta^2 = 0.04$ or greater.

H4.   The strength of the relationship between spending on physical booster packs in digital stores and problem gambling will be equivalent to magnitude $\eta^2 = 0.04$ or greater.

## Method

### Ethics

Ethical approval for this research was granted by the York St. John University Cross-Departmental Ethics Board. Informed consent was gathered from each participant.

### Design

We conducted a cross-sectional online survey with a sample of players of collectible card games aged 18+. Participants were recruited via posts on reddit (www.reddit.com), a popular internet bulletin board. Posts were made to a variety of 'subreddits', or specialist interest bulletin boards, for a variety of collectible card games. The specific subreddits that the survey was posted to are as follows: r/wixoss; r/cardfightvanguard; r/FoWtcg; r/CardWarsTCG; r/PokemonTCG; r/TCG; r/pkmntcg; r/yugioh. Posts mentioned that the survey is about "physical card games, and more specifically, spending preferences in physical card games" but does not mention loot boxes. It invites individuals to take part in "a *very short* questionnaire (5

minutes max) that asks a couple of questions about spending on card games". No reimbursement was given to participants in return for their participation in this study.

In total, 731 responses were collected. Four participants with duplicate IP addresses were removed from the sample. One participant listed their age as 100, and their gender as 'cat'. They were removed from the sample as non-serious. This left a total of 726 responses.

Participant gender was recorded via an open-response text entry box. 655 participants (90%) described themselves as 'Male' or 'M'. 49 described themselves as 'Female' or 'F' (6%). The remaining 22 participants identified as other categories (e.g., Non bigendered, genderfluid).

123 participants (16.9%) were aged 18–19; 520 (71.6%) 20–29; 70 (9.6%) 30–39, and 13 (1.8%) 40+.

The majority of the sample (n = 434) indicated that their residence was within the USA, the UK (n = 67), and Canada (n = 52). However, the sample was highly international, with 32 participants coming from Germany, 24 from Australia, and the remaining 117 participants coming from countries ranging from Italy to Indonesia.

## Measurements

**Spending on physical booster packs in real-world stores** was measured by asking participants the following question, where $CURRENCY was replaced dynamically by the currency in the participant's home country:

During the last month, how much money in $CURRENCY would you say that you have spent on booster packs in *real-world stores* (such as your local Walmart)?

Please not that we are only interested in spending on *physical* booster packs—not virtual or digital ones. Cards that you can hold in your hand.

If you have not bought a booster pack in a real-world store during the past month, just enter '0'.

**Spending on physical booster packs in digital stores** was measured by asking participants the following question, where $CURRENCY was replaced dynamically by the currency in the participant's home country:

During the last month, how much money in $CURRENCY would you say that you have spent on booster packs in *digital stores* (such as a website that sells boosters and posts them to you)?

Please note that we are only interested in spending on *physical* booster packs—not virtual or digital ones. Cards that you can hold in your hand.

If you have not bought a booster pack in a digital store during the past month, just enter '0'.

All spending data was converted into US Dollars and rank-transformed prior to analysis. Studies that investigate loot box spending (e.g. [19]) have routinely found extreme outliers in spending data. Rank transformation was therefore applied prior to all analysis, in an identical fashion to as in [16].

**Problem gambling** was measured via the Problem Gambling Severity Index [40]. This is a 9-item instrument that asks individuals a series of questions about the gambling related behaviours. For example, one item asks "Have you borrowed money or sold anything to get money to gamble?". In response to each of these items, participants are asked to select one of four options: (0) Never; (1) Sometimes; (2) Most of the time; (3) Almost always. An overall measurement of problem gambling is formed from the sum of these scores, with values ranging

from 0 to 27. These scores are then commonly transformed into diagnostic categories, with values of 0 indicating a non problem gambler; values of 1–4 indicating an individual at low risk of problem gambling; values of 5–7 representing an individual at moderate risk of problem gambling, and values of 8 or more indicating the presence of problem gambling [41]. The PGSI has demonstrated good internal validity (Cronbach's alpha > 0.8) in a variety of previous studies (e.g. [8,42–44]). In this sample, Cronbach's alpha was measured at 0.82.

### Justification of statistical analyses

This paper seeks to replicate the analyses present within Zendle and Cairns (2018). During review, some questions were raised regarding the justification of these analyses. In order to highlight the decision making regarding these analyses, we incorporate additional information below.

An anonymous reviewer asked why parametric analyses (e.g. ANOVA) were not used here, and nonparametric tests used instead. The intention here was two-fold: First, to replicate prior analyses; second, to use a test which does not require the strict distributional assumptions of an ANOVA. It is not clear that normality assumptions associated with ANOVA would be met in this case.

This reviewer also asked why data were rank-transformed prior to analysis. Again, this was done in an attempt to replicate prior work as closely as possible. One important thing to note here is that the crucial preregistered analyses here are all rank-sum tests: as such, they involve rank transformation during calculation, and thus this transformation cannot change the calculated test statistic. We appreciate that one may (rightly) view this transformation as unnecessary for some analyses, but it is also true that this transformation cannot in any way affect the statistics that these tests calculate, and hence the validity of their results.

The same reviewer also asked whether additional tests were necessary in this case in order to establish median shift via the Kruskal Wallis. This is an interesting point. We would note that the lack of a significant result in the case of all analyses means that equality of variability is not necessary for the interpretation of the Kruskal Wallis statistic (i.e., we are never in the position of trying to work out whether dominance or median shift has occurred, because the test highlights no significant differences). Again, this approach is in line with the replication of prior work.

Finally, it was asked why the PGSI was grouped in the way that was undertaken, and whether this might affect results. The PGSI measure was grouped categorically in line with common practice using this instrument: There is evidence that the diagnostic categories associated with the PGSI have important real-world meaning. We acknowledge that grouping responses by category tends to reduce the information yielded by the data, and that one might suggest that this information encodes a significant relationship between problem gambling and booster pack spending. However, crucially we would also note that we conduct an equivalence test of the correlation between 'raw' (i.e. uncategorised) problem gambling scores and both forms of booster pack spending.

### Results

A description of spending on booster physical booster packs in both real-world and digital stores, split by problem gambling severity, is given below as Table 1.

Contrary to H1, a Kruskal Wallis H Test indicated that there was no statistically significant effect of problem gambling severity on spending on physical booster packs in real-world stores, $\chi^2(3) = 6.03$, $p = 0.110$, $\eta^2 = 0.004$. A box-plot showing the amount spent on physical

**Table 1. Medians, inter-quartile ranges, and mean ranks for booster pack spending, split by problem gambling severity and spending in physical vs. online stores.**

| Problem gambling severity | Spending on physical booster packs in real-world stores | Spending on physical booster packs in digital stores | N |
|---|---|---|---|
| People without gambling problems | Median: $12.90, IQR: $39.00 Mean rank: 349.69 | Median: $0.00, IQR: $3.52 Mean rank: 365.77 | 429 |
| Low-risk gamblers | Median: $18.75, IQR: $50.00 Mean rank: 379.5 | Median: $0.00, IQR: $0.00 Mean rank: 352.21 | 244 |
| Moderate-risk gamblers | Median: $20.00, IQR: $69.00 Mean rank: 388.87 | Median: $0.00, IQR: $50.00 Mean rank: 432.00 | 35 |
| People with gambling problems | Median: $33.75, IQR: $118.75 Mean rank: 433.92 | Median: $0.00, IQR: $0.00 Mean rank: 329.31 | 18 |

booster packs in real world stores by different problem gambling severity groups is depicted below as Fig 1.

Consistent with H2, a Kruskal Wallis H Test indicated that there was a statistically significant effect of problem gambling severity on spending on spending on physical booster packs in digital stores, $\chi^2(3) = 8.54$, p = 0.035, $\eta^2 = 0.008$. In order to further explore this effect, pairwise comparisons were then conducted to measure the effects of problem gambling on booster pack spending in digital stores between all groups of gamblers via a series of 6 Mann-Whitney U tests. Bonferroni corrections were applied to the results of these tests, lowering the alpha level of the tests to 0.05/6, or 0.008. Only one significant difference was observed–low-risk gamblers spent significantly less than moderate risk gamblers, $U (n_1 = 244, n_2 = 35) = 3329.5$, p

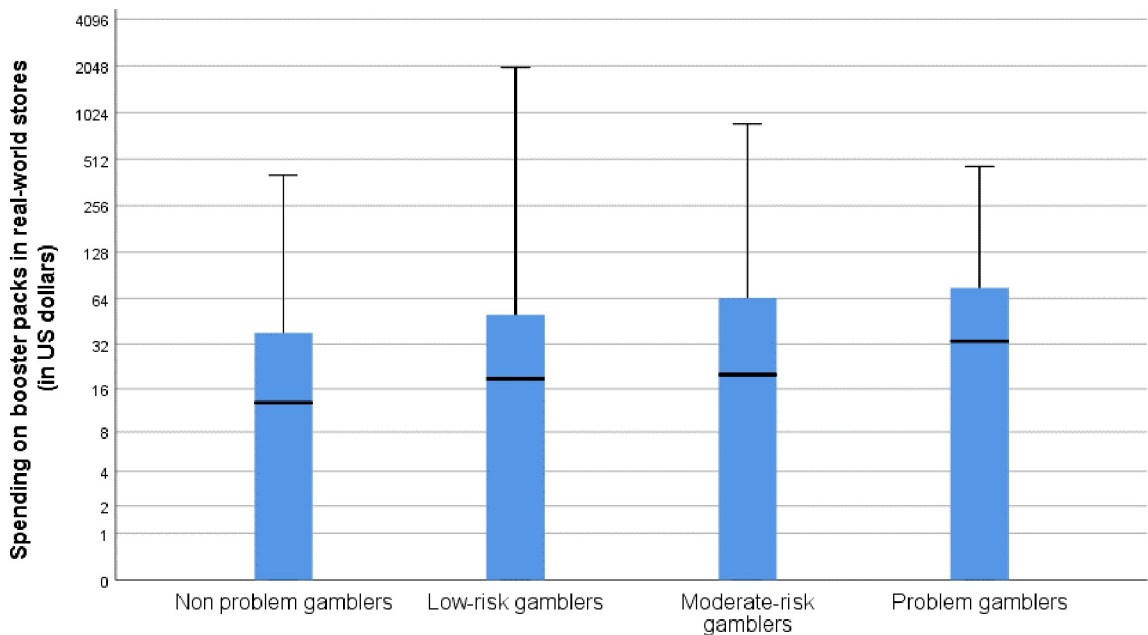

**Fig 1. Box-plot showing the relationship between problem gambling severity (people without gambling problems, low-risk gamblers, moderate-risk gamblers, people with gambling problems) and spending on booster packs in real-world stores.** The y-axis is $\log_2$ transformed in order to allow the display of extreme spending values. Bars represent medians, boxes represent quartiles, and whiskers represent range of spending.

**Table 2. Results of pairwise analysis of effects of problem gambling severity on spending on physical booster packs in digital stores.**

| Problem gambling severity | | W | $n_1$ | $n_2$ | p-value | r | Equivalent $\eta^2$ |
|---|---|---|---|---|---|---|---|
| Non people with gambling problems | Low-risk gamblers | 54292 | 429 | 244 | 0.285 | 0.041 | 0.001 |
| | Moderate-risk gamblers | 6139.5 | 429 | 35 | 0.022 | 0.106 | 0.011 |
| | People with gambling problems | 4248 | 429 | 18 | 0.347 | 0.044 | 0.001 |
| Low-risk gamblers | Moderate-risk gamblers | 3329.5 | 244 | 35 | 0.005* | 0.166 | 0.027 |
| | People with gambling problems | 2335.5 | 244 | 18 | 0.53 | 0.038 | 0.001 |
| Moderate-risk gamblers | People with gambling problems | 404 | 35 | 18 | 0.048 | 0.272 | 0.073 |

Results that are significant at the $p<0.008$ level are marked *.

= .005. All comparisons are reported below as Table 2, and a box plot showing the amount spent on physical booster packs in digital stores is depicted below as Fig 2.

To test whether the strength of the relationship between spending on physical booster packs in real-world stores and problem gambling will be equivalent to magnitude $\eta^2 = 0.04$ or greater (**H3** we employed the TOST procedure described by Lakens [45]). An equivalence test was calculated over the correlation between spending on physical booster packs in real-world stores and raw problem gambling severity scores. Equivalence bounds were placed at $r_s = 0.2$ and $r_s = -0.2$, equivalent to $\eta^2 = 0.04$ and $\eta^2 = -0.04$. Results indicated that the correlation between booster pack in real-world stores was statistically equivalent to this effect size. The 90% confidence interval for rho was calculated at 0.011–0.133 ($\eta^2$: 0.000–0.017), $p < 0.001$. To test whether the strength of the relationship between spending on physical booster packs in digital stores and problem gambling will be equivalent to magnitude $\eta^2 = 0.04$ or greater" (**H4**

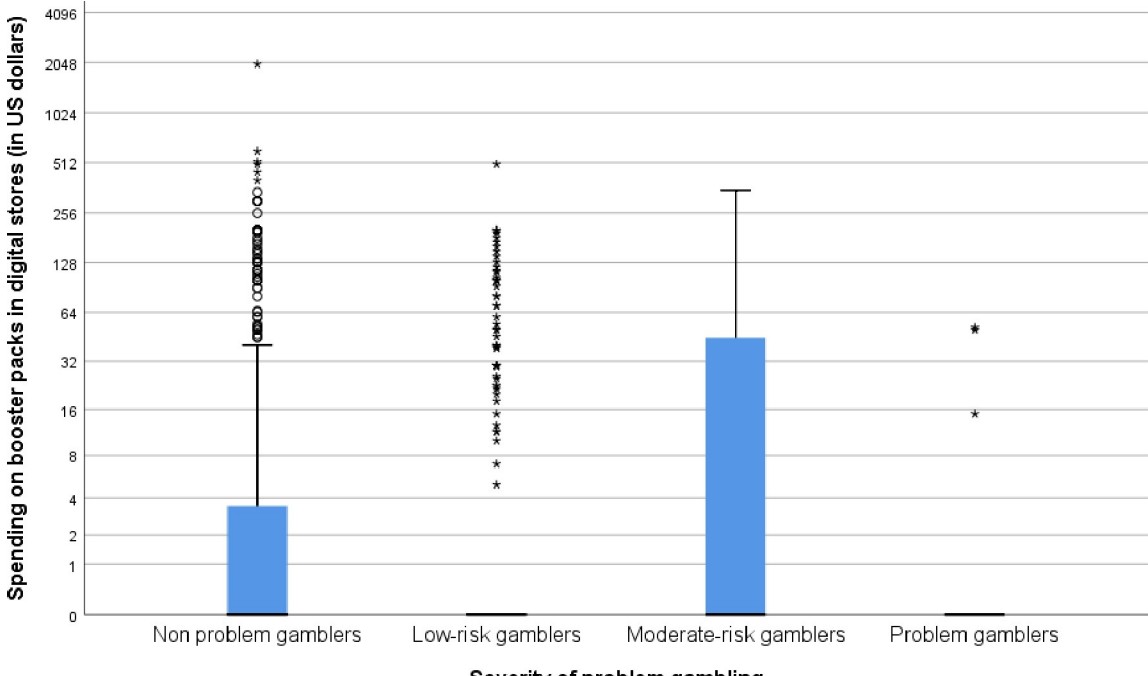

**Fig 2. Box-plot showing the relationship between problem gambling severity (non people with gambling problems, low-risk gamblers, moderate-risk gamblers, people with gambling problems) and spending on physical booster packs in digital stores.** The y-axis is $\log_2$ transformed in order to allow the display of extreme spending values. Bars represent medians, boxes represent quartiles, and whiskers represent range of spending.

the same equivalence testing procedure was employed. Results again indicated that the correlation between booster pack in real-world stores was statistically equivalent to $\eta^2 = 0.04$. The 90% confidence interval for rho was calculated at -0.061–0.061 ($\eta^2$: -0.003–0.003), $p < 0.001$.

The procedure we followed is the standard method for carrying out an equivalence test over a correlation, as defined in [45]. This procedure involves transforming a correlation coefficient to a z-score via Fisher's transformation. However, this calculation typically takes place using Pearson's *r*, not Spearman's rho. Spearman's rho is strongly related to Pearson's rho: Indeed, rho is simply a calculation of Pearson's *r* over the ranks of a dataset. Pearson's *r* and Spearman's rho therefore share many features: For example, the sampling distribution of both these statistics asymptotically approaches normality as sample size increases. However, they may also differ in important ways: For example, the standard deviation of rho may be differently distributed to the same statistic calculated over r. It is therefore unclear whether this procedure may introduce some measure of error or bias when used in this context.

In order to address this concern, bootstrap confidence intervals were calculated over both Spearman rank correlation coefficients (number of repetitions = 1000). Results indicated that the 95% bootstrap confidence interval for the relationship between spending on physical booster packs in real-world stores and problem gambling lay entirely within the equivalence bounds of -0.2 to 0.2. The 95%CI spanned -0.003 to 0.155 (equivalent $\eta^2$: 0.000–0.024). Results indicated that the 95% bootstrap confidence interval for the relationship between spending on physical booster packs in digital stores and problem gambling also lay entirely within the equivalence bounds of -0.2 and 0.2. The 95%CI spanned 0.000 and 0.075 (equivalent $\eta^2$: 0.000–0.005).

During peer review, one anonymous peer reviewer noted the small number of moderate-risk gamblers within the study (n = 35). The reviewer suggested that moderate-risk gamblers be merged with high-risk gamblers to form a single group of n = 53, and the analysis re-conducted. Accordingly, data were recoded as suggested and exploratory Kruskal-Wallis tests were run. Under this exploratory coding scheme, no significant difference was observed between groups (non problem, low-risk, moderate-risk or people with gambling problems) for either spending on physical booster packs in real-world stores ($\chi^2(2) = 5.45$, p = 0.065, $\eta^2 = 0.004$) or spending on physical booster packs in digital stores ($\chi^2(2) = 3.64$, p = 0.161, $\eta^2 = 0.002$).

## Discussion

The present project investigated whether spending on CCG booster packs differed between groups based on were related to problem gambling symptomology. Results of Kruskal-Wallis tests and follow-up Mann Whitney U tests provided little evidence for the existence of an important link between spending money on booster packs and problem gambling. Specifically, no significant relationship was observed between problem gambling severity and spending on physical booster packs in real-world stores. Similarly, whilst a statistically significant overall link was observed between spending on physical booster packs in digital stores and problem gambling (p = 0.035), the magnitude of this link was trivial in size, and failed to exceed our thresholds for clinical relevance. The starting point for effects of a small size is conventionally set at $\eta^2 = 0.01$ [46], with effects smaller than this size commonly considered trivial [39]. The effect observed here was of $\eta^2 = 0.008$.

Indeed, subsequent pairwise comparisons suggested few significant differences between gambling risk groups. The pairwise comparison of groups amongst which a difference would be suggestive of an important link between booster pack spending and problem gambling did not return statistically significant results. For example, the difference in booster pack spending

between non people with gambling problems and people with gambling problems was not statistically significant–in fact, the effect size associated with this comparison was only of magnitude $\eta^2 = 0.001$. Indeed, the only statistically significant effect observed during all pairwise comparisons was a difference in spending between low-risk gamblers and moderate-risk gamblers. However, again, the effect size associated with this difference was of only $\eta^2 = 0.016$, suggesting it was unlikely to be of practical importance.

An even stronger picture was painted by the results of equivalence tests. In order to determine the practical utility of a test's result, researchers commonly look at the effect size associated with that test. The idea that small effect sizes may indicate that a statistical result may not have significant real-world importance is discussed by Ferguson [39], a widely-cited primer on the interpretation of effect sizes. Ferguson [39] proposes that a 'suggested minimum' for effects to even be considered practically important in a clinical context be placed at $r = 0.2$ ($\eta^2 = 0.04$). An equivalence test rejected the hypothesis that the relationship between problem gambling and spending on physical booster packs in real-world stores was equal to or larger than this magnitude. Similarly, a second equivalence test rejected the hypothesis that the relationship between problem gambling and spending on physical booster packs in digital stores was of this magnitude or larger. In order to be as conservative as possible given the nonparametric nature of the analyses employed here, exploratory bootstrap confidence intervals were calculated for the magnitude of the correlation between both problem gambling and spending in real-world stores, and problem gambling and spending in digital stores. These 95% confidence intervals both fell entirely within the equivalence bounds specified above. This is particularly important given that a recent meta-analysis reveals that the association between loot box purchasing and problem gambling symptomology is $r > .20$ [47]. That loot box purchasing and problem gambling symptomology are associated with $r$'s exceeding these guidelines for clinical relevance while purchasing of physical card packs for collectable card games were not, suggests potentially important differences between these activities in terms of their risks to people with gambling problems.

Why was there no important association observed between booster pack spending and problem gambling? As we noted in our introduction, there are multiple plausible explanations for why purchases of booster packs might share a fundamentally different relationship with gambling than loot boxes. For example, the physical nature of booster packs may make purchasing and opening them significantly less quick and easy than purchasing and opening loot boxes. A number of interventions designed to limit gambling harm (e.g., Limit Setting, Drummond et al., 2019) [7] are targeted at the slowing of decision making to encourage slow deliberative processing over fast impulsive decision making [48]. One might imagine that the purchasing and opening of loot boxes can occur much faster than booster packs as one would either have to physically drive to a store to purchase the packs, or wait for physical shipping of the products if purchased online. These delays may be enough to disrupt impulsive and potentially problematic behaviour.

## Limitations

The study described above represents a first attempt to replicate effects seen amongst loot boxes in the context of physical collectible card games. Whilst it provides important evidence when it comes to this question, it is nonetheless limited in a number of important ways. In this section, we describe the primary limitations of the approach taken above, and put forward suggestions for future work that builds on it.

The most important limitation of the study described above mirrors the most important limitation of contemporary research on loot boxes: As with many studies that explore the

potential for harm present in loot boxes, this piece of research is cross-sectional in nature (e.g. [6,8,11,16,20,49,50]). It is fair to note that a causal relationship between booster pack spending and problem gambling cannot be captured via a cross-sectional approach. Evidence for a causal relation would require a longitudinal or experimental approach. Further work in this domain should build on the research conducted here using these designs.

A second key limitation of the research conducted here is the obtained sample: Our participants consisted of 726 volunteers, drawn from the online bulletin board service reddit. Samples drawn from reddit appear to skew young, and skew male (e.g. [2,51]). Indeed, this is a limitation of the work that we seek to replicate, which likewise draws its sample from reddit. A lack of academic research into players of physical collectible card games makes it unclear how representative these samples are of players of collectible card games in general. However, one may credibly propose that selecting samples of volunteers from bulletin boards designed for enthusiastic players of collectible card games may lead researchers to end up with an unrepresentatively engaged sample of players of these games. On the one hand, this is an advantage of the approach taken when it comes to ecological validity: If one is interested in spending on booster packs being linked to problem gambling, it is worthwhile looking at a sample of individuals who are likely to spend money on booster packs. However, on the other hand, one might reasonably argue that such an approach suppresses variability within our sample, and may mask the presence of real correlations within the population. Further research with more representative samples of players of collectible card games is necessary to determine which of these is the case.

Another limitation of this study is the differing timeframe between the PGSI and our measurements of spending behaviour: Whilst the PGSI measures gambling activity over the course of the previous year, we only assessed booster pack spending during the previous month, as in line with other literature (e.g. [52]). One might argue that gamers may not remember how much money they spent 10 or 11 months ago, and that capturing spending during a more proximal period may lead to more accurate measurement. However, one might also imagine a situation in which an individual spends heavily on both booster packs and gambling for 11 months, and then ceases both activities for the month prior to measurement occurring: Such behaviour would be captured adequately by the PGSI, but not by our measure of spending. Further research using more distal spending measures is necessary to determine what the consequence of this approach to measurement is.

Our sample contained a small number of individuals who gambled in a problematic manner, and a small number of moderate-risk gamblers. Our analyses may therefore be underpowered to determine differences between these groups and other groups when it comes to booster pack spending. Indeed, as noted above, further research using larger and more representative samples is needed when it comes to understanding spending on collectible card games. It is also important to note that all equivalence tests rejected the idea, even when power was increased by combining moderate and high risk groups (ps <0.001), presenting strong evidence that the observed relationship was equivalent to one of a clinically significant magnitude.

An additional limitation to our approach relates to our strict replication of the techniques used in an early paper on loot box spending [2]. It was our intention to replicate this study as closely as possible, in order to minimise any additional factors that might be responsible for an observed difference in results between our paper and the study that we were attempting to replicate. However, the study that we replicated, and by extension, the present study did not involve extensive data cleaning: For example, very fast responses were not purged from either study. One may reasonably suggest that replicating this approach may lead to issues with data quality. This is a legitimate criticism: In fact, the authors of the original paper that is replicated

here later replicated both their analyses (and results) with such procedures in place [20]. As such, further work should focus on replicating the results obtained here with attention checks and greater data cleaning.

A final limitation of our employed approach relates to loot box spending. The purpose of this study was to attempt to address questions regarding the relationship between booster pack spending and gambling problems. As such, we neither measured nor analysed data regarding loot box purchasing. However, during review it was noted that augmenting our analyses with data on loot box purchasing might give a more holistic and rounded picture of how individuals spent money on randomised rewards, and the inter-relationship between this factor and gambling problems. Indeed, one could imagine a situation in which this factor were measured, and results indicated that (within a single sample), loot box spending was linked to gambling problems, whilst booster pack spending were not. Further work that adopts such a holistic approach is recommended in order to both replicate and increase confidence in the findings observed here.

## Conclusions

Our results suggest that spending money on booster packs does not appear to be linked to problem gambling in the same way that spending money on loot boxes is. Loot boxes have repeatedly been linked to problem gambling symptoms (e.g., [7–9,16,52]. In contrast, these results show that CCGs are not linked to problem gambling in the same way. Two clear corollaries are evident from these findings 1) individual examination of the potentially harmful effects of activities that resemble gambling are required to determine which activities may require legislation, and 2) that loot boxes share some features with other unregulated activities should not be taken to suggest that loot boxes do not require regulation. Loot boxes have consistently been linked to problem gambling, whereas CCGs do not appear to similarly relate to increased spending for players with gambling problems. Investigation of other activities with similar structural features to loot boxes and traditional forms of gambling may be valuable to understand whether there are other activities which users with gambling problems spend more on than peers without gambling problems.

## Acknowledgments

No further parties contributed to this study but did not meet criteria for authorship.

## Author Contributions

**Conceptualization:** David Zendle, Rachel Meyer.

**Data curation:** David Zendle, Rachel Meyer.

**Formal analysis:** David Zendle, Lukasz Walasek, Paul Cairns.

**Investigation:** David Zendle, Rachel Meyer.

**Methodology:** David Zendle.

**Project administration:** David Zendle, Rachel Meyer.

**Resources:** David Zendle.

**Supervision:** David Zendle.

**Validation:** David Zendle.

**Visualization:** David Zendle.

**Writing – original draft:** David Zendle.

**Writing – review & editing:** David Zendle, Lukasz Walasek, Paul Cairns, Aaron Drummond.

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
