## [Decision Letter · Decision Letter 0]

22 Dec 2020

PONE-D-20-32484

Links between problem gambling and spending on booster packs in collectible card games: A conceptual replication of research on loot boxes

PLOS ONE

Dear Dr. Zendle,

Thank you for submitting your manuscript to PLOS ONE. After careful consideration, we feel that it has merit but does not fully meet PLOS ONE’s publication criteria as it currently stands. Therefore, we invite you to submit a revised version of the manuscript that addresses the points raised during the review process.

Two Reviewers evaluated the manuscript and suggested maior revisions, focusing in particular on methodological aspects and clarification of background and hypotheses. 

To their detailed revisions I would add just a comment on CCGs: it could be important to notice that, when one is deeply engaged in some card game, they end up possessing a significant portion of their decks or cards by means different from buying. You can win cards, packs or event entire decks (e.g., Keyforge) by participating to regular store tournaments, or have exchanges with other players that could be more or less advantageous (e.g., entire composite decks in exchange of one particularly valuable card). Moreover, packs or decks could be easily received as a gift when participating to official events. This could be useful to understand the difference between CCGs and gambling. 

We look forward to receiving your revised manuscript.

Kind regards,

Stefano Triberti, Ph.D.

Academic Editor

PLOS ONE

Journal Requirements:

2. Please provide additional details regarding participant consent.

In the ethics statement in the Methods and online submission information, please ensure that you have specified what type you obtained (for instance, written or verbal, and if verbal, how it was documented and witnessed).

If your study included minors, state whether you obtained consent from parents or guardians.

If the need for consent was waived by the ethics committee, please include this information.

Reviewers' comments:

Reviewer's Responses to Questions

**Comments to the Author**

1. Is the manuscript technically sound, and do the data support the conclusions?

Reviewer #1: Partly

Reviewer #2: Partly

2. Has the statistical analysis been performed appropriately and rigorously? 

Reviewer #1: Yes

Reviewer #2: No

3. Have the authors made all data underlying the findings in their manuscript fully available?

Reviewer #1: Yes

Reviewer #2: Yes

4. Is the manuscript presented in an intelligible fashion and written in standard English?

Reviewer #1: Yes

Reviewer #2: No

5. Review Comments to the Author

Reviewer #1: Thank you for the opportunity to review the present manuscript examining the relationship between collectible card games and problem gambling. The manuscript was well-written and investigates a timely research question given the comparison of loot boxes to trading cards. I believe the study adds valuable insight into the rapidly growing literature on the convergence of gaming-gambling. My main concern is the small sample size of people in the moderate-risk to problem gambling categories. I note my further comment below.

1. I would suggest using person centered language such as people with gambling problems rather to reduce the stigma associated with problem gambling.

2. On page 13, the authors state that “Whilst booster packs may be bought either in a physical store or via an online storefront, they must always be opened in the real world.” I believe this is not entirely accurate. Taking the MAGIC the Gathering example, Magic Online and Magic Arena allow booster packs to be open virtually. Thus, the paragraph on key distinction between loot boxes and CCGs may need to be amended.

3. Related to above, some of the concerns raised of loot boxes that are different compared to CCGs are: (i) questionable advertising practices targeting vulnerable individuals, and (ii) not knowing the odds of loot boxes, and or the ease in which the odds can be manipulated. These concerns may add context to the difference between loot boxes and CCGs and provide further argument for there potential differences. These may also help explain why there were no observed relationship.

4. Very minor comment, but the text for data availability and ethics are single spaced.

5. Were participants compensated for the study?

6. I am curious if the authors took any measures to screen for data quality such as attention checks, checking unusually fast attention checks? In other words, how are the authors sure that the responses gathered were of high quality?

7. Did the authors also assess past year spending on CCGs? If not, this should be mentioned as a limitation given that the PGSI measures past 12 months, while spending was measured in the past year. For example, it may be possible that some participants may have begun to limit their spending on CCGs in the past month, which would not be captured and may attenuated the relationship between PGSI and CCGs

8. I would also be curious if the authors assessed loot box purchases as well. I believe looking at the relationship between loot boxes, CCGs, and problem gambling would provide a more holistic picture.

9. The low sample sizes in the moderate-risk and problem gambling category may account for the null effects. Was a power analyses conducted to examine sample size to detect the clinically meaningful effect? I would like to see results of combining the moderate-risk and problem categories to boost the Ns of the category.

10. The authors discuss clinically meaningful effect. I believe the discussion section would be strengthened if the authors speak to the effect size in the relationship between loot boxes and problem gambling severity, which I believe are generally above .04. I believe this would help strengthen the argument that CCGs do not pose similar risks of problem gambling as loot boxes.

11. The limitation section is lacking (non-existent). Overall, I really enjoyed the paper but it is limited in some ways:

- Differing time frames between PGSI and CCG spending behavior

- Highly selected and convenience sample from Reddit, and as such results may not

generalize to CCG players

- Did not (?) assess loot boxes, which would have provided a more complete picture.

- Cross-sectional. Perhaps a clinically meaningful effect would be observed longitudinally?

- Small sample size of people in the moderate risk and problem gambling categories.

Reviewer #2: The study examines the relationship between collectable card game (CCG) booster pack purchasing, in the real world and online, and problem gambling symptom severity. A rationale for examining this relationship is that there is a positive relationship for loot box purchasing and problem gambling which is suggested to guide public policy decisions for such activities. It was hypothesized that because CCGs are similar to loot boxes, then there would be a significant relationship between categories of problem gambling severity and CCG pack purchasing. The study sought to answer this question using online surveys that were completed by a cross-sectional sample of 731 participants recruited from “Reddit.” It was ultimately concluded that there is no relationship between problem gambling severity and spending on booster packs. Specifically, there was no relationship between purchasing of physical booster packs and gambling severity and only a weak relationship between virtual booster pack purchasing and gambling severity (which failed to meet the threshold selected by the researchers). The study was interesting and seeks to answer an important question that could impact policy decisions surrounding the regulation of activities that resemble gambling. However, there are several limitations. Please take the following comments into consideration:

1. Introduction: An issue with the manuscript in the current form is whether the information presented in the introduction leads, conceptually, to the hypotheses that are initially tested. Based on my reading of the introduction, the reasoning, and how the two activities are contrasted, it seemed clear that CCG booster packs are not the same as loot boxes and, therefore, there should be no relationship between problem gambling and CCG booster purchasing (such an assumption would require equivalence testing, which the authors do, but I wonder why this approach wasn’t selected from the outset). Though some similarities between loot boxes and CCGs are discussed, many more differences are presented and the arguments seems stronger that the two activities are not the same. Thus, it is as if there is a disconnect between the conceptual framework offered in the introduction and the initial hypotheses the study seeks to test. To give an example, in the first paragraph, the statement challenges the case that CCGs are a form of gambling (i.e., "just because something appears to resemble gambling…") and statements such as these reoccurs throughout the introduction, but in the last paragraph the similarities between CCG boosters and loot boxes are evoked to justify that there should be a connection between this activity and problem gambling. Please consider changing the tone or rewording sections of the introduction to provide more congruence.

2. Pg. 3, Ln. 4: This sentence does not read well. Perhaps this is a syntax error or typographical error?

3. Methods: For the PGSI, please report previous reliabilities from other studies that used the PGSI as well as how the instrument performed in the current study (this is especially important in this context because the sample was international and questionnaires were collected fully online).

4. Pg. 10, Ln.17: The fact that one of the groups contains a small number of female participants would need to be evaluated given that there are differences between women and men in how they process and regulate the effects of problem gambling. Also, are there gender differences that should be addressed in terms of Reddit users or gambling in general? If so, please mention in the methods section or, if it cannot be addressed, as a study limitation in the discussion section.

5. Methods: Please consider including more information about recruitment of participants via Reddit.

6. Methods: The data analytic plan should be described in greater detail including the decisions for the analyses that are conducted. Why was the Kruskal Wallis H Test used rather than ANOVA? The problem of outliers is mentioned and a transformation was used to correct for this, but then why the use of a nonparametric test which is robust to outliers? Why not use the untransformed values then? Also, were all assumptions for the Kruskal Wallis H Test met (e.g., equal variability)? What was the rationale for using the PGSI measure categorically rather than continuously in this study? In short, there appear to be many statistical decisions and interpretations made by the authors that aren’t explained or justified to the reader.

7. Discussion: Some of the statements made are far too strong given the exploratory nature and limitations of this study and should not be generalized to all unregulated gambling-like activities (e.g., “Arguments that loot boxes are equivalent to other unregulated activities that resemble gambling appear to be empirically unjustified.”)

8. Discussion: There are a number of limitations to this study, but none are mentioned. Please include a through discussion of the limitations. Also, please consider including recommendations for future research to build upon and expand the results of the current study (e.g., an experimental or longitudinal design?).

9. Overall: The manuscript should undergo additional proofreading. There are misplaced commas, extra spaces, and some peculiar formatting choices (e.g., double parentheses for citations, authors' names referred to directly in sentence but still placed in parentheses, references without page numbers or only the starting page). As I believe this journal uses Vancouver style, which I am less familiar with than APA, perhaps I am simply unfamiliar with such conventions, but I would recommend reviewing the potential issues I mention here to be sure. Finally, the formatting for some of the graphs and figures made it difficult to follow the data.

6. PLOS authors have the option to publish the peer review history of their article (what does this mean?). If published, this will include your full peer review and any attached files.

Reviewer #1: No

Reviewer #2: No

---

## [Author Response · Author response to Decision Letter 0]

4 Feb 2021

Dear Professor Triberti and the editorial team,

Thank you for the opportunity to revise this manuscript “Links between problem gambling and spending on booster packs in collectible card games: A conceptual replication of research on loot boxes.” (PONE-D-20-32484). We wish to thank the editor and reviewers for their detailed comments. We provide a point-by-point response to all issues that were raised below (in bold).

We believe that the reviewer comments have substantially improved the manuscript. 

We look forward to hearing from you in due course,

Kind regards,

Editor Comments

You said: “Two Reviewers evaluated the manuscript and suggested maior revisions, focusing in particular on methodological aspects and clarification of background and hypotheses. 

To their detailed revisions I would add just a comment on CCGs: it could be important to notice that, when one is deeply engaged in some card game, they end up possessing a significant portion of their decks or cards by means different from buying. You can win cards, packs or event entire decks (e.g., Keyforge) by participating to regular store tournaments, or have exchanges with other players that could be more or less advantageous (e.g., entire composite decks in exchange of one particularly valuable card). Moreover, packs or decks could be easily received as a gift when participating to official events. This could be useful to understand the difference between CCGs and gambling.” 

Our response: This is an excellent point, and one that you are right in stating we do not mention. We have extended our introduction to briefly touch on this topic, though we were unable to find any academic sources that discuss it. Specifically on page 10, we now state: “Furthermore, it is important to note that the way that cards are played and collected amongst players of physical collectible card games may differ from how loot box contents are accrued by players of video games. For instance, anecdotal reports suggest that players of collectible card games may end up owning large number of cards through means other than buying. For example, the tangible nature of physical card games mean that players may gift or trade cards with each other. In a video game, rules for the transfer of property between players are controlled by the architects of the game, and may not allow such transfers to take place.: For example, loot box contents in the video game Last Shelter: Survival are locked to a player’s account, and may not be traded between players (Ballou et al., 2020). Furthermore, winning packs, cards, or decks by participating in formal and informal tournaments is thought to be a common occurrence amongst players of physical card games.: Such bonuses may not translate to all loot box implementations.”

Reviewer 1 Comments

You said: “Thank you for the opportunity to review the present manuscript examining the relationship between collectible card games and problem gambling. The manuscript was well-written and investigates a timely research question given the comparison of loot boxes to trading cards. I believe the study adds valuable insight into the rapidly growing literature on the convergence of gaming-gambling. My main concern is the small sample size of people in the moderate-risk to problem gambling categories. I note my further comment below.

1. I would suggest using person centered language such as people with gambling problems rather to reduce the stigma associated with problem gambling.”

Our response: We thank the reviewer and agree with their suggestion. The manuscript has been redrafted in such a way that person-centred language is used throughout.

You said: “2. On page 13, the authors state that “Whilst booster packs may be bought either in a physical store or via an online storefront, they must always be opened in the real world.” I believe this is not entirely accurate. Taking the MAGIC the Gathering example, Magic Online and Magic Arena allow booster packs to be open virtually. Thus, the paragraph on key distinction between loot boxes and CCGs may need to be amended.”

Our response: This is a really good point, and shows a key area in which we must clarify our manuscript: Magic Arena is a video game in which individuals spend real-world money in order to obtain a randomised digital reward. Functionally, there is no difference between Magic Arena’s booster packs and -say- player packs in FIFA. Under the definitions used in this manuscript, Magic Online and Magic Arena would therefore be selling ‘loot boxes’ rather than ‘booster packs’. The key distinction we make between these forms is not what things look like, but whether they are physical or digital: Hearthstone (a video game) does not sell booster packs. We did not gather data from players of video games for this project, but only from people who bought physical booster packs. The questions within our survey were framed in such a way as to make this clear. We have augmented our manuscript both to make this key point clear; and to highlight the distinction between booster packs and loot boxes outlined above. Specifically, we now note that: “Indeed, some similarities between physical card collections and digital loot boxes are so great that some video game loot boxes take the appearance of ‘virtual’ card packs. For example, FIFA Ultimate Team’s ‘player packs’ are presented in-game in a way that mirrors the opening of physical packs of playing cards. Even more strikingly, some video games adopt overarching in-game metaphors that directly mimic traditional physical collectible card games, and incorporate loot boxes into these games in the form of digital card packs. Hearthstone, for example, is a computer game in which players collect and battle using virtual cards in a similar manner to physical card games such as Magic: The Gathering and the Pokemon collectible card games. Players of Hearthstone may pay real world money to purchase virtual ‘packs’ of digital cards. However, crucially, this monetisation technique involves disbursing virtual and not physical goods, and takes place within the context of a video game: whilst it may have graphics which seek to mimic real-world booster packs, it nonetheless constitutes a video game loot box under our definitions, due to its virtual content and presence within a video game rather than a booster pack which distributes physical cards for a physical card game. 

The similarities outlined above have led to comparisons between collectible card games and loot boxes by members of the video game industry (Taylor, 2018). These similarities have led to significant policy interest in whether the effects of loot boxes and booster packs are fundamentally equivalent: The terms of reference of a recent evidence call by the UK government, for example, requested information regarding “Whether any harms identified [to be associated with loot boxes] also apply to offline equivalents of chance mechanisms, such as buying packs of trading cards.”(DCMS, 2020).” (p.7).

You said: “3. Related to above, some of the concerns raised of loot boxes that are different compared to CCGs are: (i) questionable advertising practices targeting vulnerable individuals, and (ii) not knowing the odds of loot boxes, and or the ease in which the odds can be manipulated. These concerns may add context to the difference between loot boxes and CCGs and provide further argument for there potential differences. These may also help explain why there were no observed relationship”.

Our response: This is a salient point, and we agree that the manuscript could be enhanced by taking it into account. We have augmented both our discussion and introduction in order to take it into consideration. Specifically on page 10 we now state: “Finally, it is important to note that specific technological differences between loot boxes and booster packs may be responsible for different relationships between these things and problem gambling. For example, many video games do not disclose the odds of loot boxes, and sometimes even algorithmically adjust the odds of receiving different rewards as people spend money (Ballou et al., 2020): This kind of manipulation is not possible when buying booster packs. In a similar vein, aggressive or otherwise misleading advertising regarding microtransactions are reported by players of video games (Petrovskaya & Zendle, 2021): Such advertising may differ in its form and prevalence within the context of physical collectible card games.”

You said: “4. Very minor comment, but the text for data availability and ethics are single spaced.”

Our response: We have adjusted the formatting to make sure that these are double-spaced.

You said: “5. Were participants compensated for the study?”

Our response: Participants receive no compensation for participation. We have added this detail to the manuscript. “No reimbursement was given to participants in return for their participation in this study” (p.12).

You said: “6. I am curious if the authors took any measures to screen for data quality such as attention checks, checking unusually fast attention checks? In other words, how are the authors sure that the responses gathered were of high quality?”

Our response: This is an interesting point. Our focus here was to replicate a previous paper. That paper did not apply any such cut-offs, so we did not put any in place. We have described this as a limitation of the approach taken. “An additional limitation to our approach relates to our strict replication of the techniques used in an early paper on loot box spending (Zendle & Cairns, 2018a). It was our intention to replicate this study as closely as possible, in order to minimise any additional factors that might be responsible for an observed difference in results between our paper and the study that we were attempting to replicate. However, the study that we replicated, and by extension, the present study did not involve extensive data cleansing: For example, very fast responses were not purged from either study. One may reasonably suggest that replicating this approach may lead to issues with data quality. This is a legitimate criticism: In fact, the authors of the original paper that is replicated here later replicated both their analyses (and results) with such procedures in place (Zendle & Cairns, 2019). As such, further work should focus on replicating the results obtained here with attention checks and greater data cleaning.” (Page 24).

You said: “7. Did the authors also assess past year spending on CCGs? If not, this should be mentioned as a limitation given that the PGSI measures past 12 months, while spending was measured in the past year. For example, it may be possible that some participants may have begun to limit their spending on CCGs in the past month, which would not be captured and may attenuated the relationship between PGSI and CCGs”

“8. I would also be curious if the authors assessed loot box purchases as well. I believe looking at the relationship between loot boxes, CCGs, and problem gambling would provide a more holistic picture.”

Our response: These are both salient points. We assessed neither of these things during this paper. We agree with the reviewer that these are interesting analyses, and have expanded our discussion to highlight them as topics for future research. Specifically on page 23-24 of the revised manuscript we note “Another limitation of this study is the differing timeframe between the PGSI and our measurements of spending behaviour: Whilst the PGSI measures gambling activity over the course of the previous year, we only assessed booster pack spending during the previous month, as in line with other literature (e.g. (Zendle, 2019a)). One might argue that gamers may not remember how much money they spent 10 or 11 months ago, and that capturing spending during a more proximal period may lead to more accurate measurement. However, one might also imagine a situation in which an individual spends heavily on both booster packs and gambling for 11 months, and then ceases both activities for the month prior to measurement occurring: Such behaviour would be captured adequately by the PGSI, but not by our measure of spending. Further research using more distal spending measures is necessary to determine what the consequence of this approach to measurement is.” And on page 

9. The low sample sizes in the moderate-risk and problem gambling category may account for the null effects. Was a power analyses conducted to examine sample size to detect the clinically meaningful effect? I would like to see results of combining the moderate-risk and problem categories to boost the Ns of the category.

Our response: As suggested by the reviewer, we have combined these categories and repeated our analyses, taking care to flag these new analyses as exploratory. In addition, we would note that we calculated an equivalence test over the correlation coefficient between raw problem gambling severity scores and both forms of booster pack spending. No effect was again observed here. Specifically on page 20 of the revised manuscript we note: “During peer review, one anonymous peer reviewer noted the small number of moderate-risk gamblers within the study (n=35). The reviewer suggested that moderate-risk gamblers be merged with high-risk gamblers to form a single group of n=53, and the analysis re-conducted. Accordingly, data were recoded as suggested and exploratory Kruskal-Wallis tests were run. Under this exploratory coding scheme, no significant difference was observed between groups (non problem, low-risk, moderate or high risk) for either spending on physical booster packs in real-world stores (χ2(2) = 5.45, p=0.065, η2 = 0.004) or spending on physical booster packs in digital stores (χ2(2) = 3.64, p=0.161, η2 = 0.002).”

10. The authors discuss clinically meaningful effect. I believe the discussion section would be strengthened if the authors speak to the effect size in the relationship between loot boxes and problem gambling severity, which I believe are generally above .04. I believe this would help strengthen the argument that CCGs do not pose similar risks of problem gambling as loot boxes.

Our response: This is an excellent point and clearly something that we have missed from our discussion. We have extended it to take this point into account. On page 21 we now elaborate that “In order to determine the practical utility of a test’s result, researchers commonly look at the effect size associated with that test. The idea that small effect sizes may indicate that a statistical result may not have significant real-world importance is discussed in (Ferguson, 2009), a widely-cited primer on the interpretation of effect sizes. (Ferguson, 2009) proposes that a ‘suggested minimum’ for effects to even be considered practically important in a clinical context be placed at r = 0.2” and on page 22 we note “This is particularly important given that a recent meta-analysis reveals that the association between loot box purchasing and problem gambling symptomology is r > .20 (Garea et al., 2020). That loot box purchasing and problem gambling symptomology are associated with r’s exceeding these guidelines for clinical relevance while purchasing of physical card packs for collectable card games were not, reveals important differences between these activities in terms of their risks to people with gambling problems.”

You said: “11. The limitation section is lacking (non-existent). Overall, I really enjoyed the paper but it is limited in some ways:

- Differing time frames between PGSI and CCG spending behavior

- Highly selected and convenience sample from Reddit, and as such results may not

generalize to CCG players

- Did not (?) assess loot boxes, which would have provided a more complete picture.

- Cross-sectional. Perhaps a clinically meaningful effect would be observed longitudinally?

- Small sample size of people in the moderate risk and problem gambling categories.”

Our response: We agree with the limitations defined by the reviewer, and regret not making them clearer in the manuscript. In response to this holistic point, we have created a separate ‘Limitations’ subsection within our discussion (pp. 22-25). We have used this section to discuss each of the points raised by the reviewer, and have also used this section to raise some additional points with reference to the limitations of this study.

Reviewer 2 Comments

You said: “The study examines the relationship between collectable card game (CCG) booster pack purchasing, in the real world and online, and problem gambling symptom severity. A rationale for examining this relationship is that there is a positive relationship for loot box purchasing and problem gambling which is suggested to guide public policy decisions for such activities. It was hypothesized that because CCGs are similar to loot boxes, then there would be a significant relationship between categories of problem gambling severity and CCG pack purchasing. The study sought to answer this question using online surveys that were completed by a cross-sectional sample of 731 participants recruited from “Reddit.” It was ultimately concluded that there is no relationship between problem gambling severity and spending on booster packs. Specifically, there was no relationship between purchasing of physical booster packs and gambling severity and only a weak relationship between virtual booster pack purchasing and gambling severity (which failed to meet the threshold selected by the researchers). The study was interesting and seeks to answer an important question that could impact policy decisions surrounding the regulation of activities that resemble gambling. However, there are several limitations. Please take the following comments into consideration:

1. Introduction: An issue with the manuscript in the current form is whether the information presented in the introduction leads, conceptually, to the hypotheses that are initially tested. Based on my reading of the introduction, the reasoning, and how the two activities are contrasted, it seemed clear that CCG booster packs are not the same as loot boxes and, therefore, there should be no relationship between problem gambling and CCG booster purchasing (such an assumption would require equivalence testing, which the authors do, but I wonder why this approach wasn’t selected from the outset). Though some similarities between loot boxes and CCGs are discussed, many more differences are presented and the arguments seems stronger that the two activities are not the same. Thus, it is as if there is a disconnect between the conceptual framework offered in the introduction and the initial hypotheses the study seeks to test. To give an example, in the first paragraph, the statement challenges the case that CCGs are a form of gambling (i.e., "just because something appears to resemble gambling…") and statements such as these reoccurs throughout the introduction, but in the last paragraph the similarities between CCG boosters and loot boxes are evoked to justify that there should be a connection between this activity and problem gambling. Please consider changing the tone or rewording sections of the introduction to provide more congruence.”

Our response: This point is really well-taken. You are right: We are trying to walk a thin line here, and possibly not treading it perfectly. I think that the place where we fell down was highlighting the existence of two competing narratives: There is an argument to be made that loot boxes are the same as collectible card games; there is an argument to be made that they are fundamentally different to collectible card games. This has led to governmental stakeholders being unsure whether these things are different or the same, which may have important consequences. We have adjusted our introduction to describe this more clearly, and hope that our adjusted introduction is better at motivating the research presented here. For instance we now note on page 7 of the revised manuscript: “The similarities outlined above have led to comparisons between collectible card games and loot boxes by members of the video game industry (Taylor, 2018). These similarities have led to significant policy interest in whether the effects of loot boxes and booster packs are fundamentally equivalent: The terms of reference of a recent evidence call by the UK government, for example, requested information regarding “Whether any harms identified [to be associated with loot boxes] also apply to offline equivalents of chance mechanisms, such as buying packs of trading cards.”(DCMS, 2020).”

You said: “2. Pg. 3, Ln. 4: This sentence does not read well. Perhaps this is a syntax error or typographical error?”

Our response: Thanks for catching this. We have rephrased the statement and hopefully it now reads more clearly.

You said: “3. Methods: For the PGSI, please report previous reliabilities from other studies that used the PGSI as well as how the instrument performed in the current study (this is especially important in this context because the sample was international and questionnaires were collected fully online).”

Our response: We have augmented the manuscript with details about PGSI reliability in previous studies, and in this study. Specifically on page 15-16 of the revised manuscript we state: The PGSI has demonstrated good internal validity (Cronbach’s alpha > 0.8) in a variety of previous studies (e.g. (Barbaranelli et al., 2013; Bertossa et al., 2014; Drummond, Sauer, Ferguson, et al., 2020; Orford et al., 2010)). In this sample, Cronbach’s alpha was measured at 0.82.”

You said: “4. Pg. 10, Ln.17: The fact that one of the groups contains a small number of female participants would need to be evaluated given that there are differences between women and men in how they process and regulate the effects of problem gambling. Also, are there gender differences that should be addressed in terms of Reddit users or gambling in general? If so, please mention in the methods section or, if it cannot be addressed, as a study limitation in the discussion section.”

Our response: Reddit samples appear to be overwhelmingly young and male. This is a limitation of the approach (and a limitation of the work we seek to replicate). We have augmented our newly-created limitations subsection to discuss the generalisability of our sample in detail. On page 23-24 we now note that “A second key limitation of the research conducted here is the obtained sample: Our participants consisted of 726 volunteers, drawn from the online bulletin board service reddit. Samples drawn from reddit appear to skew young, and skew male (e.g. (Petrovskaya & Zendle, 2020; Zendle & Cairns, 2018a)). Indeed, this is a limitation of the work that we seek to replicate, which likewise draws its sample from reddit. A lack of academic research into players of physical collectible card games makes it unclear how representative these samples are of players of collectible card games in general. However, one may credibly propose that selecting samples of volunteers from bulletin boards designed for enthusiastic players of collectible card games may lead researchers to end up with an unrepresentatively engaged sample of players of these games. On the one hand, this is an advantage of the approach taken when it comes to ecological validity: If one is interested in spending on booster packs being linked to problem gambling, it is worthwhile looking at a sample of individuals who are likely to spend money on booster packs. However, on the other hand, one might reasonably argue that such an approach suppresses variability within our sample, and may mask the presence of real correlations within the population. Further research with more representative samples of players of collectible card games is necessary to determine which of these is the case.”

You said: “5. Methods: Please consider including more information about recruitment of participants via Reddit.”

Our response: We have augmented our manuscript to include significantly more detail about recruitment via Reddit. On page 12-23 we now state: “The specific subreddits that the survey was posted to are as follows: r/wixoss; r/cardfightvanguard; r/FoWtcg; r/CardWarsTCG; r/PokemonTCG; r/TCG; r/pkmntcg; r/yugioh. Posts mentioned that the survey is about “physical card games, and more specifically, spending preferences in physical card games” but does not mention loot boxes. It invites individuals to take part in “a *very short* questionnaire (5 minutes max) that asks a couple of questions about spending on card games”. No reimbursement was given to participants in return for their participation in this study.” 

You said: “6. Methods: The data analytic plan should be described in greater detail including the decisions for the analyses that are conducted. Why was the Kruskal Wallis H Test used rather than ANOVA? The problem of outliers is mentioned and a transformation was used to correct for this, but then why the use of a nonparametric test which is robust to outliers? Why not use the untransformed values then? Also, were all assumptions for the Kruskal Wallis H Test met (e.g., equal variability)? What was the rationale for using the PGSI measure categorically rather than continuously in this study? In short, there appear to be many statistical decisions and interpretations made by the authors that aren’t explained or justified to the reader.”

Our response: These are all sound points. They all stem from one key source: This paper attempts to replicate a prior ‘loot box’ paper, but in the domain of booster packs; our analyses were conducted to mirror the analyses within that paper. With that said, we would suggest that our analyses still make statistical and conceptual sense. 

However, we appreciate that we have perhaps failed to communicate this within our manuscript – your objections are well-taken! We have tried to augment the manuscript to make it a bit more clear why we did what we did, with specific detail added to a ‘Justification of statistical analyses’ subsection within our method. This subsection contains the following specific content:

“This paper seeks to replicate the analyses present within Zendle and Cairns (2018). During review, some questions were raised regarding the justification of these analyses. In order to highlight the decision making regarding these analyses, we incorporate additional information below.

An anonymous reviewer asked why parametric analyses (e.g. ANOVA) were not used here, and Kruskal Wallis tests used instead. The intention here was two-fold: First, to replicate prior analyses; second, to use a test which does not require the strict distributional assumptions of an ANOVA. It is not clear that normality assumptions associated with ANOVA would be met in this case. 

This reviewer also asked why data were rank-transformed prior to analysis. Again, this was done in an attempt to replicate prior work as closely as possible. One important thing to note here is that the crucial preregistered analyses here are all rank-sum tests: as such, they involve rank transformation during calculation, and thus this transformation does not change the calculated test statistic. We appreciate that one may (rightly) view it as unnecessary for some analyses, but it is also true that this transformation cannot in any way affect the statistics that these tests calculate, and hence the validity of their results. 

The same reviewer also asked whether additional tests were necessary in this case in order to establish median shift via the Kruskal Wallis. This is an interesting point. We would note that the lack of a significant result in the case of all analyses means that equality of variability is not necessary for the interpretation of the Kruskal Wallis statistic (i.e., we are never in the position of trying to work out whether dominance or median shift has occurred, because the test highlights no significant differences). 

Finally, it was asked why the PGSI was grouped in the way that was undertaken, and whether this might affect results. The PGSI measure was grouped categorically in line with common practice using this instrument: There is evidence that the diagnostic categories associated with the PGSI have important real-world meaning. We acknowledge that grouping responses by category tends to reduce the information yielded by the data, and that one might suggest that this information encodes a significant relationship between problem gambling and booster pack spending. However, crucially we would also note that we conduct an equivalence test of the correlation between ‘raw’ (i.e. uncategorised) problem gambling scores and both forms of booster pack spending: These tests both suggest a relationship between these variables that is equivalent to zero.”In addition to this, we would note that when we have grouped participants in the alternative fashion proposed by Reviewer 1, the analysis remained nonsignificant. We believe that these points highlight the robustness of our results to alternative statistical approaches.

You said: “7. Discussion: Some of the statements made are far too strong given the exploratory nature and limitations of this study and should not be generalized to all unregulated gambling-like activities (e.g., “Arguments that loot boxes are equivalent to other unregulated activities that resemble gambling appear to be empirically unjustified.”)”

Our response: We thank the reviewer for this comment and have removed this statement.

You said: “8. Discussion: There are a number of limitations to this study, but none are mentioned. Please include a through discussion of the limitations. Also, please consider including recommendations for future research to build upon and expand the results of the current study (e.g., an experimental or longitudinal design?).”

Our response: In response to both this feedback, and the suggestions of Reviewer 1, we have augmented the manuscript with an additional ‘Limitations’ subsection (pages 22-25 of the revised manuscript).

You said: “9. Overall: The manuscript should undergo additional proofreading. There are misplaced commas, extra spaces, and some peculiar formatting choices (e.g., double parentheses for citations, authors' names referred to directly in sentence but still placed in parentheses, references without page numbers or only the starting page). As I believe this journal uses Vancouver style, which I am less familiar with than APA, perhaps I am simply unfamiliar with such conventions, but I would recommend reviewing the potential issues I mention here to be sure. Finally, the formatting for some of the graphs and figures made it difficult to follow the data.”

Our response: We have undertaken a thorough proofread of the manuscript and corrected a number of small errors. We hope that we have addressed the concerns that you describe here.

---

## [Decision Letter · Decision Letter 1]

15 Feb 2021

Links between problem gambling and spending on booster packs in collectible card games: A conceptual replication of research on loot boxes

PONE-D-20-32484R1

Dear Dr. Zendle,

We’re pleased to inform you that your manuscript has been judged scientifically suitable for publication and will be formally accepted for publication once it meets all outstanding technical requirements.

Kind regards,

Stefano Triberti, Ph.D.

Academic Editor

PLOS ONE

Additional Editor Comments (optional):

Reviewers' comments:

Reviewer's Responses to Questions

**Comments to the Author**

1. If the authors have adequately addressed your comments raised in a previous round of review and you feel that this manuscript is now acceptable for publication, you may indicate that here to bypass the “Comments to the Author” section, enter your conflict of interest statement in the “Confidential to Editor” section, and submit your "Accept" recommendation.

Reviewer #1: All comments have been addressed

2. Is the manuscript technically sound, and do the data support the conclusions?

Reviewer #1: Yes

3. Has the statistical analysis been performed appropriately and rigorously? 

Reviewer #1: Yes

4. Have the authors made all data underlying the findings in their manuscript fully available?

Reviewer #1: Yes

5. Is the manuscript presented in an intelligible fashion and written in standard English?

Reviewer #1: Yes

6. Review Comments to the Author

Reviewer #1: I would like to thank the authors for the careful consideration given to my suggestions in the previous reviews. I believe the authors have addressed all my comments and feel they are submitting a stronger paper for consideration. I wish the authors the best of luck with their work and look forward to seeing the paper in print!

7. PLOS authors have the option to publish the peer review history of their article (what does this mean?). If published, this will include your full peer review and any attached files.

Reviewer #1: No

---

## [Editor Report · Acceptance letter]

11 Mar 2021

PONE-D-20-32484R1 

Links between problem gambling and spending on booster packs in collectible card games: A conceptual replication of research on loot boxes 

Dear Dr. Zendle:

I'm pleased to inform you that your manuscript has been deemed suitable for publication in PLOS ONE. Congratulations! Your manuscript is now with our production department. 

Kind regards, 

on behalf of

Dr. Stefano Triberti 

Academic Editor

PLOS ONE